# Biological Mechanisms of *S*-Nitrosothiol Formation and Degradation: How Is Specificity of *S*-Nitrosylation Achieved?

**DOI:** 10.3390/antiox10071111

**Published:** 2021-07-12

**Authors:** Christopher M. Massa, Ziping Liu, Sheryse Taylor, Ashley P. Pettit, Marena N. Stakheyeva, Elena Korotkova, Valentina Popova, Elena N. Atochina-Vasserman, Andrew J. Gow

**Affiliations:** 1Department of Pharmacology & Toxicology, Ernest Mario School of Pharmacy, Rutgers University, Piscataway, NJ 08848, USA; christopher.massa@gmail.com (C.M.M.); zl140@scarletmail.rutgers.edu (Z.L.); shery56@gmail.com (S.T.); ashleyppettit@gmail.com (A.P.P.); 2RASA Center in Tomsk, Tomsk Polytechnic University, 634050 Tomsk, Russia; stakheyevam@oncology.tomsk.ru (M.N.S.); atochina@pennmedicine.upenn.edu (E.N.A.-V.); 3Institute of Natural Resources, Tomsk Polytechnic University, Lenin Av. 30, 634050 Tomsk, Russia; eikor@mail.ru (E.K.); vap25@tpu.ru (V.P.); 4Perelman School of Medicine, University of Pennsylvania, Philadelphia, PA 19104, USA

**Keywords:** nitrosothiol, nitric oxide, cysteine, post-translational modification, thiol

## Abstract

The modification of protein cysteine residues underlies some of the diverse biological functions of nitric oxide (NO) in physiology and disease. The formation of stable nitrosothiols occurs under biologically relevant conditions and time scales. However, the factors that determine the selective nature of this modification remain poorly understood, making it difficult to predict thiol targets and thus construct informatics networks. In this review, the biological chemistry of NO will be considered within the context of nitrosothiol formation and degradation whilst considering how specificity is achieved in this important post-translational modification. Since nitrosothiol formation requires a formal one-electron oxidation, a classification of reaction mechanisms is proposed regarding which species undergoes electron abstraction: NO, thiol or S-NO radical intermediate. Relevant kinetic, thermodynamic and mechanistic considerations will be examined and the impact of sources of NO and the chemical nature of potential reaction targets is also discussed.

## 1. Introduction

Nitric oxide (NO) is a pluripotent signaling molecule produced by almost every organism and utilized in over 46 different physiological functions, especially within the cardiovascular system [1,2,3]. The post-translational modification of protein cysteine residues by NO to form S-nitrosothiols (SNO) has emerged as an essential physiological signaling mechanism [4]. Such modifications occur through a process entitled S-nitrosylation (or nitrosation as it has also been termed), which satisfies all criteria as a mechanism for the regulation of cell physiology. There are a number of mechanisms that have been proposed to explain the formation of SNOs [5] and it has proven difficult to define which and how individual SNOs are formed within particular proteins. It appears that protein S-nitrosylation is a highly regulated process, which operates within narrow spatiotemporal constraints [6]. Nitrosothiols are, by their very nature, not permanent, and some investigators suggest that SNOs are merely a staging post on route to disulfide bond formation [7]. Stoichiometrically, the authors revealed an increased RSSR-to-RSNO ratio in the smooth muscle cell proteome after treatment with a nitric oxide donor (CysNO) for most cysteine residues identified using mass spectrometry [8]. However, the contribution of either disulfide or nitrosylation to downstream pathways was not quantified nor was the generalizability of NO’s function to any strong leaving group established.

Refined proteomic techniques have identified a range of proteins that are nitrosylated in vivo [9,10], and databases of SNO-modified cysteines are available [11]. SNO formation has been demonstrated to occur over biologically relevant time scales in physiologic intracellular concentration ranges [12]. Protein SNOs are produced on demand in response to specific stimuli resulting in a change in target function [13]. SNO formation has been shown to exert both activating and inactivating roles in biological pathways through numerous mechanisms, including suppression, activation and the allosteric modulation of enzyme function [14], the initiation of apoptosis [15], the promotion/suppression of proteosomal degradation [15,16], and the regulation of transcription factor activity [17,18] as well as transcriptional regulation through the alteration of DNA methylation [19] and histone acetylation [20]. SNO’s essential signaling roles are ubiquitous across kingdoms, having been shown to participate in such diverse signaling such as bacterial transcription [17], plant immunology [21], mammalian nervous transmission [22] and both mitochondrial-dependent and -independent apoptotic pathways [15,23]. The termination of the SNO effect by denitrosylation has been shown to readily occur in vivo [24,25]. Despite the large number of cysteine residues within the cells, S-nitrosylation is a highly selective process, with a paucity of protein thiols demonstrating evidence of modification in vivo.

The complexity of SNO signaling can be appreciated by considering the effects of S-nitrosylation within cellular pathways. Nowhere is this more evident than in considering cell death processes, which can be both activated and inactivated by NO via S-nitrosylation [26]. The anti-apoptotic protein Bcl-2 is regulated by its degradation following ubiquitination, a process that is opposed by S-nitrosylation [27], leading to the NO-mediated promotion of cell survival. In contrast, the nitrosylation of key regulatory proteins can promote cell death, notably via inhibiting the E3 ubiquitin ligase function of Parkin [28] and the anticaspase-3 function of XIAP. The caspases themselves are targets for nitrosylation with the active site cysteine being a target for modification, which promotes cell survival [29]. Even a brief consideration of the complexity of SNO-based signaling within this one cellular pathway demonstrates how the system is finely tuned and emphasizes the need to understand the regulatory forces in effect.

Past review articles have highlighted the great breadth of functional consequences of NO-mediated modification in both physiology and pharmacology [30,31,32,33,34,35,36]. In addition, the general chemistry of SNO biology has been considered in the context of proteomic investigations of SNOs found in vivo [37], while there have been multiple papers published on quantifying the production of nitrosothiols [38,39,40,41,42]. However, how the chemical mechanisms of SNO formation and degradation lead to the targeted specificity that is observed in biological systems has been less emphasized. In examining these chemical mechanisms, several key factors emerge that influence the likelihood of modifying specific cysteine residues under different cellular conditions. Interplay between sources of NO, potential intracellular targets for reaction, essential chemistry of SNO formation and destruction, and the nature of protein/thiol targets gives rise to the rich diversity of SNOs found in physiological systems and give provide insight into potential mechanisms by which pathology arises. The areas to be covered in this review are outlined in the accompanying figure and are focused on understanding how the specificity of SNO signaling can be achieved (Figure 1).

## 2. Sources of NO

The primary source of NO within biological systems is the three isoforms of the enzyme nitric oxide synthase; eNOS, iNOS and nNOS. The structure function and mechanisms of regulation of these enzymes have been reviewed previously [1,43,44,45]. In addition to the enzymatic oxidation of arginine by NOS isoforms, NO may also be generated in biological systems through the reduction of nitrite. Nitrite reductase activity is a key mechanism by which NO can be generated during hypoxia, leading to mitochondrial regulation [46]. Several mechanisms for nitrite reduction with potential relevance to mammalian physiology have been proposed: acidification of nitrite, molybdenum enzyme catalysis, and heme-iron catalysis. Dietary nitrates are found in vegetables such as leafy greens and beetroots and can be converted into nitrite using nitrate reductases within commensal oral bacteria, which use nitrate as a terminal electron acceptor to generate ATP. Newly formed nitrite is swallowed with saliva and enters the digestive tract [47]. Nitrite within the low pH environment of the stomach exists in its protonated form, HNO_2_ [48]. The bimolecular reaction of 2 HNO_2_ results in the disproportionate production of water and N_2_O_3_, which can in turn act as a direct nitrosating agent or decompose via homolytic cleavage at the N–N bond to produce NO and NO_2_. The low pH of the gastric contents is demonstrated as essential for this mechanism, as pre-treatment with proton pump inhibitors abolishes NO production [48,49]. The presence of additional reducing agents within the gut such as polyphenols and ascorbate further promote the formation of NO from dietary nitrates. It should be noted that the significance of dietary sources of NO equivalents in SNO formation has been demonstrated in humans, where SNO levels in the plasma and red blood cells can be increased 4-to-10-fold by beet juice ingestion [50]. At neutral pH, nitrite reduction has been observed under markedly hypoxic conditions, wherein oxygen fails to act as an electron acceptor for molybdenum containing enzymes [51], such as Xanthine oxidoreductase and aldehyde dehydrogenase. In hypoxia, the electron flow pathway through the FADH, and the four FeS clusters and the molybdenum ion is altered, facilitating the reduction in NO_2_^−^ to NO [52,53]. 

Hemoproteins, such as hemoglobin, have also been suggested as capable of nitrite reduction in low O_2_ conditions. In vitro studies have demonstrated that pentacoordinate heme reduces HNO_2_ to NO and OH^−^ at physiologic pH and low oxygen tensions, oxidizing the heme from ferrous to ferric iron. For many hemoproteins, this reaction is competitive with O_2_ binding, and the reaction of oxy-heme may result in nitrate production, making nitrite reduction feasible with greater degrees of hypoxia [54,55,56], consistent with physiological responses, such as vasodilation, at low oxygen tensions. Such chemistry is believed to underlie ischemic preconditioning, as myoglobin-deficient mice exposed to cardiac ischemia-reperfusion fail to experience the protective effect of nitrite and have a significant diminution of NO formation, as assessed by electron paramagnetic resonance [57]. Appreciable NO production through nitrite reduction by hemoproteins appears more speculative, as the high concentration of hemoglobin in red cells is kinetically predicted to scavenge produced NO [58], while the reaction of nitrite with oxyhemoglobin favors nitrate production [59]. In addition, NO, nitrite, and peroxynitrite have been found to be substrates of the heme peroxidases, potentially limiting NO bioavailability [60,61,62]. Hemoproteins may also contribute to the NO pool by catalyzing the production of nitrosating species N_2_O_3_ from nitrite [63,64] as well as direct nitrite reduction [65]. Heme-based catalytic activity has also been observed for eNOS [66,67], mitochondrial electron transport chain constituents [68,69], cytochrome P450 isoforms [70,71] and neuroglobin [72], imparting differential tissue production and regulation based on basal protein levels, signaling which favors pentacoordinated over hexacoordinated heme [72,73] and tissue pO2 [66]. As some disagreement between kinetic predictions, in vitro experiments and animal studies exists in the literature, the relevance of the above mechanisms to physiology and pathology, as well as the conditions which favor their contribution, remain topics of investigation [74]. It should be noted that the non-metal center reduction of nitrite can occur, as in the case of UVA exposure to the skin that can generate NO from nitrite [75].

## 3. Reactive Targets of NO

Nitric oxide has a unique redox potential that allows it to readily both accept and donate electrons within the physiological milieu [76,77]. In short, this means that NO can participate in a wide range of redox reactions as well as radical recombination events; this leads to a wide range of potential reactions for NO-related species within the biological milieu [78]. The nature of specific reaction targets may increase, diminish or otherwise alter the subsequent transfer of the NO moiety. The redox reactions generate a diversity of nitrogen species including the NO radical, NO^+^, NO^−^ and higher oxides, such as N_2_O_3_. These species are often grouped under the umbrella term reactive nitrogen species, or RNS; however, these individual entities have different chemistries, which influences the propensity and mechanism by which they are involved in S-nitrosylation. Here, we consider several reactive targets of NO, emphasizing the potential consequences on the formation of protein SNOs: namely reactive oxygen species, metals and thiols.

### 3.1. Oxygen Species

Within an anaerobic aqueous environment, NO is inherently stable, though it reacts readily under aerobic conditions [79]. The autooxidation of NO with O_2_ is a third order reaction that generates 2 NO_2_. This NO_2_ product may undergo radical–radical reaction with NO to form N_2_O_3_, which reacts readily with water to produce 2 NO_2_^−^ + 2 H^+^. In hydrophobic environments, N_2_O_3_ is capable of decomposition into NO^+^ and NO_2_^−^ [80,81]. Though chemically plausible, the reaction rates generating these species are likely quite low in the physiologic range of NO concentrations [82]. NOS activation with concurrent oxidative stress—as occurs with metabolic alteration or the activation xanthine oxidase or NADPH oxidase—may result in the formation of a variety of reactive oxides. In the presence of O_2_^−^, NO reacts to form the highly reactive peroxynitrite anion [83]. In addition to its varied roles in pathology, peroxynitrite can facilitate SNO formation [84], although the mechanism maybe indirect [85]. In reaction with hydroxyl radical, NO acts as an antioxidant, producing NO_2_^−^, though this reaction is presumably quite rare, due to hydroxyl radical’s high reactivity [86]. The generation of higher order nitrogen oxides may alter the capacity for nitrosylation of different biological targets. In the context of pathologic ROS production, the modifications of targets may occur, potentially via the formation of thiyl radicals, which are not regulated by SNO during normal physiology. The activity of iNOS under conditions of oxidative stress has the most significant potential to generate peroxynitrite [87], particularly when GSH is depleted or in the absence of superoxide dismutase. 

### 3.2. Metals

NO reacts readily with redox active metals both within proteins and as free cations, such as unbound iron. These reactions lead to one of three outcomes: (1) the formation of a stable metal–nitrosyl complex; (2) transient nitrosothiol bond leading to eventual thiol oxidation/disulfide bond formation; or (3) formation of a stable nitrosothiol facilitated by the metal. 

Copper- and iron-containing proteins such as soluble guanylate cyclase, aconitase, ferritin, plastocyanin, and cytochrome c oxidase demonstrate formation of metal–nitrosyl complexes. The chemistry of these nitrosyl species is principally determined by the nature of the highest orbital occupied in the bonded structure. Thus, the coordination and redox state of the interacting metal is critical in determining the character of the bound nitrosyl [88], and nitrosyls formed with Cu^1+^ proteins containing type 3 copper centers result in a complex that favors N_2_O formation, while Cu^2+^ proteins containing type 3 or type 1 copper centers favor the production of nitrite [89]. In proteins such as cytochrome c oxidase that are capable of forming both types of nitrosyl, the outcome depends on the redox status of the protein [90,91,92]. For the purposes of this review, it is key to note that copper–nitrosyls are capable of generating nitrosating equivalents that can contribute to SNO formation, such as in the case of ceruloplasmin [93].

Similarly, NO is capable of reactions with iron in various oxidation states. The most well-known interaction for NO with iron centers is the five-coordinate binding with the heme center of guanylate cyclase [94] or deoxygenated hemoglobin [95] (although these binding reactions are straight forward [96,97]). As with copper, the redox fate of NO is dictated by the redox and coordination status of the heme iron [98]. NO is capable of reacting with both with Fe^3+^ and Fe^2+^; however, direct binding to Fe^2+^ is the most facile [99]. Proteins containing iron sulfur clusters, such as aconitase and heme peroxidases, are also capable of forming iron–nitrosyl complexes that may lead to temporary inactivation. For aconitase, formation of an iron–nitrosyl bond disrupts the 4Fe–4S cluster as it is temporarily reduced to 3Fe–4S [100]. The inactivation of this enzyme requires the breakdown of this cluster, which requires a direct nitrosation event involving a nitrosonium cation donor. Disruption of the cluster results in aconitase inactivation and the inhibition of its function as an iron-responsive element binding protein. 

Activity with Fe–S clusters and other redox metals is not limited to free NO, but also NO that has reacted with iron and thiol to form a dinitrosyl iron complex (DNIC). DNICs are formed through the bonding of two molecules of NO to ferrous iron with various thiol ligands [101]. DNICs can be formed with low molecular weight thiols or with protein thiols [102]. Reaction with low molecular weight thiols leads to the generation of a potent nitrosating agent, while protein-based DNICs are more stable [101,103]. Redox interchange between the two NO moieties leads to the production of a nitrosonium equivalent (nitrite under aqueous conditions) and a nitroxyl ion, imparting increased reactivity with additional thiols and other reactive species. In this way, low molecular weight DNICs can operate like a form of nitrosylase as they offer longer-term storage and transport for NO, while their breakdown provides nitrosating species. They are capable of crossing membranes and have been detected in cells that do not produce NO [104], where they can initiate a cascade of nitrosothiol formation and iron recycling, leading to changes in protein function and cellular signaling [105]. 

The reactivity of NO with redox active vitamins, such as B12, is not often considered when thinking of cellular targets. Various enzymes depend on the redox activity of the cobalt contained in B12 for their function. NO is capable of reacting with cobalamin^2+^ (Cbl^2+^) at physiological pH, and Cbl^3+^ at low pH [106]. However, the affinity of NO for iron and heme-containing proteins is higher than that of cobalt and Cbl. Further dissociation rates for Cbl–nitrosyl bonds are much faster than that of iron–nitrosyl bonds; however, this remains a potentially physiologically relevant reaction for NO. This pathway is potentially relevant as B12 activity has been found to attenuate the physiological effects of NO, possibly serving as an NO trap [107]. However, NOCbl readily reacts with O_2_, oxidized to NO_2_Cbl and limiting the overall contribution of this pathway [108]. 

### 3.3. Thiols

Among the most abundant targets in biological systems are cysteine thiol groups in proteins and peptides, underscoring the need for reaction specificity. The most prevalent outcomes of interaction between NO donors and reduced thiols are the formation of SNO or disulfides [109]. The NO radical is uncharged and preferentially partitions into hydrophobic compartments, increasing its local concentration within the membrane, which may influence its chemistry [80]. Though plasma membranes represent approximately 3% of the total cell volume, the significant increase in the rate of reaction between NO and molecular oxygen (30 [81] to 300 [110] fold) is suggestive that generation of these reaction products (including the potent nitrosating agent N_2_O_3_) is predominantly localized to hydrophobic regions. This “molecular lens” theory [81] has also been suggested to apply to the hydrophobic core of proteins in enabling NO autooxidation to higher oxides. Given the low concentrations of NO within the cell, the hydrophobic concentrating effect may impart feasibility to N_2_O_3_ mediated nitrosation, but at a rate insufficient to compete with radical reactions under many circumstances. Indeed, it has been demonstrated that under specific conditions, the presence of protein can reduce the yield of nitrosylation of glutathione [111]. In addition to NO and autoxidation products, other nitrogen oxides including nitrosonium equivalents, such as nitrite, and nitroxyl ions, as well as NO_2,_ are capable of reacting with free thiols [78]. In reactions between reduced thiol and NO_2_, the NO_2_ may oxidize the thiol, producing NO_2_^−^ and thiyl radical [112]. The multitude of reactive nitrogen-oxygen species suggests that thiol residues within biology do not behave identically.

Thiol groups on free cysteine amino acids have a pKa value of 8.3, suggesting less than 10% of extracellular and 5% of intracellular cysteine should exist in the thiolate ion (S-) form at a given time. Within the context of the tertiary structure of proteins, the microenvironment surrounding the cysteine may dramatically alter the likelihood of thiol deprotonation. For example, flanking charged amino acid residues proximal to a thiol can increase the stability of the deprotonated form. Conversely, cysteine residues buried within the hydrophobic core of a protein will be unstable in the charged state and thus are predicted to remain protonated. As a polar moiety, the thiol group, though preferred to the thiolate ion in hydrophobic protein regions, is less favored than the relatively nonpolar NO adduct, increasing the likelihood of S-nitrosothiol formation, provided the product is sterically permissible. The potential surface or interior localization may be suggested upon examination of the primary sequence of residues flanking a potentially modified cysteine residue [113]. It is important to consider that the coordination of charges may occur at the level of the tertiary structure, and so precise structure determination may play a key role in understanding thiol chemistry as it relates to the favorability of S-nitrosylation [114]. In some circumstances, SNO formation induces a protein conformational change with important consequences on SNO stability through charge coordination or altered steric interaction [115]. The fact that the microenvironment of a thiol residue can alter its likelihood to be protonated and the stability of any modified products implies certain residues within the entire biological system are more readily nitrosated, i.e., one should be able to observe the evidence of directed reactivity with NO. 

## 4. Specificity of SNO Formation

Only a minority of cysteine residues appear capable of participating in S-nitrosylation in the presence of an exogenous NO donor [113], with even fewer observed in vivo [116]. Despite the observation that surface accessible cysteine residues are widely found on proteins, only a small subset is believed to have functional consequences in physiologic or pathologic conditions of NO production. The skeletal muscle ryanodine receptor-1 exemplifies this point, as it possesses 100 cysteine residues, nine of which have been demonstrated to be physiologically nitrosylated in vivo and three of which are modifiable following treatment with exogenous redox-active substance [117]. Notably, reactive oxygen species permit the modification of additional cysteine residues, as opposed to abrogating nitrosylation, underscoring the complexity of thiol redox regulation. Given the vast array of potential targets for modification by NO in vivo, such exquisite selectivity—especially from a readily diffusible mediator—is noteworthy and suggests that mechanisms of the S-nitrosylation of specific targets are highly regulated. For example, although fully capable of diffusing throughout the cell, eNOS-generated NO remains near the Golgi apparatus [118]. Additionally, detected modifications were restricted to the golgi and the nucleus of these cells. The high local concentration of NO may provide even more selectivity as to which proteins are nitrosylated, and under which circumstances if spatial constraints also exist in the activity of enzymatically generated NO. 

Structural analyses of endogenously nitrosylated proteins have principally examined surrounding residues in the primary structure. This approach has been useful in relating regional hydropathy to the prediction of nitrosylation targets, however, it is likely that tertiary structure plays an essential role in determining plausibility, reaction rates and the stability of SNO formation. Since tertiary structure is a function of the entire primary sequence, information potentially important in determining tertiary structure and ultimately, the “nitrosylatability” of a cysteine thiol is lost if only a small window of primary sequence surrounding the cysteine in question is observed. All current learning algorithms using primary sequence as the dependent variable exhibit this weakness, as the incorporation of entire sequences is difficult without extensive computation times or the lossy conversion of data. 

A few motifs for nitrosylation have been identified, although none of have been found to be a consistent for all identified SNO sites or in predicting SNO formation. These motifs include the acid-base motif and hydrophobic motif which have been the most discussed, and other features have been identified such as the high exposure of the sulfur atom. However, the analysis of endogenously modified proteins found in vivo revealed that hydropathy scores, pKa values, and the accessibility of the S atom of modified cysteine residues did not differ significantly from those that were unmodified. However, endogenous nitrosylations were less likely to occur in proteins with a coiled secondary structure and there was the increased occurrence of charged resides near the modified thiol [119]. A study using 55 proteins identified in vitro showed that the same motifs did not work well for distinguishing modified proteins. Here, a more reliable acid-base motif was described as “slightly modified” compared to the original motif. This modification is a positively charged residue occurring within 6 Å and a negative charge within 8 Å. The authors also found an increased occurrence of charged flanking residues [120]. In contrast to unpublished data from our lab which observed the preservation of the CXXC motif upon examination of endogenously nitrosylated WT lungs, they found that cysteine was highly unlikely to occur at +/- 3 positions, though both found cysteine unlikely to occur at other positions as well. In addition to the differences between in vivo and in vitro protein identification, additional challenges to finding motifs in nitric oxide modification exist as the SNO proteome can be changed by disease state, phenotype, and cellular redox status. Changes in SNO pools due to the phenotype does not make it far to reason that SNPs and additional genetic variances may create varied SNO pools, further complicating the discovery of consistent motifs that are directly relevant to humans. Greater success in identifying motifs for S-nitrosylation may be found in increased the identification of proteins in vivo, separating proteins by the potential mechanism by which they would be modified with special attention to the microenvironment as it would determine what is chemically permissible. Some answers to the questions of which proteins, and when, may lie in the how and possibly the result of the modification (activation/deactivation). Marino and colleagues also suggested that additional clustering such as protein function (transferases, oxioreductases, etc.) or pathway involvement may provide more definitive answers [120]. The use of refining clusters as a means of understanding SNO motifs may be especially useful as the thiol pool for NO modification is fluid, but the evolutionary conservation of motifs may occur for proteins of a shared function. However, for this to be a viable option, greater efficiency for identifying thiol modifications in vivo is necessary and required experimentally. SNO formation in vivo is dictated by chemistry and so is the experimental detection of SNO. This point has been recently highlighted by work completed by Chung et al. revealing distinct SNOed protein targets using two different methods of SNO detection on wild-type heart samples, with minimal overlap [121]. 

NO species are also in competition with a variety of other thiol-modifying substrates, notably reactive oxygen species, nearby protein thiols, organo–metallic compounds, and glutathione. In this context, it is important to remember that cysteine residues are the targets of other NO-derived reactive molecules, such as nitrated fatty acids, via Michael addition [122]. Protein thiols, under conditions of oxidative stress, may be reversibly modified by the addition of oxygen species or the formation of disulfide bonds; however, NO, in the radical non-charged form, is not capable of producing SNO via reaction with thiols which have undergone oxidation in the excess of 1e^−^ loss. The dynamic alteration of intracellular redox state through pro-oxidant stresses, or the actions of glutathione, protein-disulfide isomerases and thioredoxin has been extensively reviewed [123,124,125], with such thiol modification changing the available pool of NO targets. For example, Xanthine oxidase produces oxidative stress in the setting of heart failure, resulting in ryanodine receptor-2 oxidation and reduction in receptor nitrosylation to sub-physiologic levels, with the consequent impairment of function [126].

In summary, there is clear evidence of specificity as only a small proportion of the potentially modifiable thiols have been found to from SNOs. However, the rules governing this specificity are clearly complex as one can see from the difficulty in defining a consensus sequence. Two factors that may be contributing to this difficulty are (1) that thiol chemistry is determined by the local environment and currently primary information is insufficient to determine tertiary structure; and (2) there are multiple mechanisms for SNO formation and decay and thus there are actually different pools of SNO. The fact that we cannot fully define the rules governing SNO selectivity is no reason to doubt that such rules exist merely so that there is more work to be done to understand this complex process, including the mechanisms governing SNO formation.

## 5. Computational Prediction

Since there is clear evidence of specificity within the population of nitrosylated proteins, it seems logical that there is a possibility to determine features that control this specificity and therefore generate a predictive methodology using computational approaches. Every modeling approach is defined by the transformation of an input vector, which is assigned to a sample of interest, to an output vector, which is then compared to the sample’s desired output vector. The individual elements of the input vector constitute the list of independent variables deemed to be correlated with the dependent variable, but do not share correlations with other elements within the list. The most basic variable, and one that is universally used in nitrosylation prediction, is the amino acids that occupy some arbitrary position relative to the central cysteine. The limit on the number of positions observed to the left or right of the cysteine that has been traditionally considered is 10. As of yet, there has not been a systematic examination of the effects of using different numbers of variables for different length peptides within the application of predicting nitrosylation sites. 

Other structural features can be derived from the amino acid composition, whether through experimental data or as the output of some defined function. AAIndex lists several physicochemical properties for the canonical 20 amino acids but effects of the proximity of other amino acids within the microenvironment are not considered. The wealth of information provided by AAIndex allows for modularity in feature selection; Li et al. used AAIndex values for polarity, secondary structure, molecular volume, codon diversity, and electrostatic charge [127], but the SNO site analyzed all properties provided by AAIndex [128]. Certain properties of individual residues, such as surface accessibility, are heavily dependent on a protein’s tertiary structure. Since the availability of three-dimensional structure information is limited, some algorithms utilize functions that provide estimations of these three-dimensional properties when given a primary sequence. Li et al. [127] calculated the disorder from the VSL2 algorithm, SNOPred calculated the amino acid disorder from an algorithm by Peng et al., and secondary structural properties (i.e., which is the amino acid part of a helix or stand structure or buried or exposed) from SCRATCH, while PSNO calculated the secondary structure properties from the PSIPRED algorithm. Substitution matrixes, which quantify the likelihood of substitution for an amino acid given the amino acid to be substituted, is also a common feature. BLOSUM was used in the development of SNO GPS, and PSI-BLAST was used in the development of SNOPred, PSNO [127]. The frequency of amino acid at their various positions can be derived from a training set, but a potential weakness is the size of the training set available. iSNO-PseAAC compared the frequency matrix for the population of sequences with cysteines positive for nitrosylation against the population of sequences with cysteines negative for nitrosylation [129]. A similar concept was incorporated into iSNO-AAPair [130], but pairwise frequencies were used for adjacent and every-other amino acid combination. There may exist pairwise frequencies with higher dependence than adjacency or every-other ordering to the outcome of nitrosylation, but exhaustive testing for even small training sets would be computationally expensive.

Feature selection is not often necessary if the feature pool is sufficiently small. GPS-SNO [131] does not require systemic feature selection, since only the BLOSUM matrix was used as input; iSNO-PseAAC [129] and iSNO-AAPair [130] only use the frequency of amino acids as input. In other scenarios, algorithms must balance between underfitting, resulting from the exclusion of features highly dependent upon the classification, and overfitting, resulting from the inclusion of features independent from the classification of interest. A possible mechanism for selecting features is to use a statistical test to quantify the discrepancy of a feature between the nitrosylated and non-nitrosylated populations. The SNO site uses the F-score to quantify the intergroup to intragroup variability of physicochemical features extracted from AAIndex at each position [132]. Another approach is to maximize the mutual information provided by a set of features while minimizing the number of informational elements needed. This concept is called the maximum relevance, minimum redundancy approach mRMR [133]. This is similar to ranking features by relative entropy, as with the Kullback–Leibler divergence described in PSNO [134]. Incremental feature selection takes an alternate approach of introducing features one at a time from a ranked list into the feature subset, then comparing the new subset against the previous subset with some measurement of performance accuracy, generally Mathew’s correlation coefficient (MCC), when trained on a regression model. The iterative process is stopped when the new subset fails to provide increases in performance, and the classification algorithm of choice is trained on the previous feature subset. This approach is used by PSNO [134] and the algorithm defined by Li et al. [133]. While the approach is subject to the issue of excluding features that may be more predictive but are lower ranked, it is a practical alternative to computing the performance of all possible feature subsets. 

Predictive model performance is generally represented by Matthew’s correlation coefficient (MCC), which is the difference between the product of true positives and true negatives and the product of false positives and false negatives, normalized to a value between −1 and 1 [135]. An MCC value of 1 would indicate that the classification technique is perfectly predictive, while an MCC value of −1 would indicate that the classification technique is perfectly unpredictive (all positives are classified as negatives and vice versa). The MCC holds advantages over accuracy (the sum of true positives and negatives divided by the total sample size) for binary classification. If an algorithm classifies all samples to be of one class regardless of the input vector, accuracy might still be high if the training set is unbalanced but MCC will approach zero. This is the case for nitrosylation prediction, where the population of nitrosylated cysteines is outnumbered by the population of cysteines negative for nitrosylation. The problem is compounded by the optimization of cost equations within iterative learning algorithms, which only attempts to minimize the sum of the differences of the output vector to the expected outputs. One possibility is to fix the specificity to some arbitrary number and adjust the coefficients of the algorithm to maximize sensitivity (i.e., the current iteration cannot decrease the specificity from the previous iteration). This was the approach used in SNO-GPS when introducing coefficients to adjust for the position and incrementing/decrementing the values of the BLOSUM matrix.

In assessing the predictive value of each web-based tool, difficulties arise as some are trained and tested on unique datasets. There is also inherent publication bias for tools developed on the same dataset, as only more predictive algorithms for the common dataset would be published in the literature as time progresses. The most popular benchmark by far is the dataset used to train GPS-SNO [131] and its associated web-based tool. The original paper uses the GPS 3.0 algorithm and benchmarks included the GPS 2.0 algorithm (which was used in phosphorylation prediction) and PSSM. The dataset consists of 504 verified sites of S-nitrosylation in 327 proteins (taken from the literature), and the unverified cysteine sites are taken as negative data. A high threshold (for the prediction of nitrosylation) settings resulted in the lowest MCC (0.1897), whereas the medium and low threshold settings resulted in MCC of 0.2175 and 0.2864, respectively. This may be explained by the relatively poor sensitivity of the high threshold setting (0.2520), so marginal increases in sensitivity drastically improved MCC. SNOSite [128] was trained on 586 S-nitrosylated sites in 384 proteins, extracted from SNAP/L-cysteine stimulated mouse endothelial cells. The testing set for SNOSite was the training set for GPS-SNO. SNOSite had modest improvements in MCC over the low-threshold setting of GPS-SNO (0.294 vs. 0.2864), but SNOSite was more sensitive (0.805 vs. 0.5357) and less specific (0.593 vs. 0.8014) than GPS-SNO. This is most likely an artifact from MDD-clustering prediction, as only one motif needs to be matched to be considered positive by the algorithm. Since the population of cysteines negative for nitrosylation will most likely be larger than the population of cysteines positive for nitrosylation, high sensitivity and low specificity models will be less accurate than models with high specificity. The iSNO-PseAAC [129] was trained on 438 proteins randomly selected from dbSNO, which resulted in 731 positive cysteine residues. For the negative cysteine residues, 810 cysteine sites were randomly selected from the 438 proteins. After training, the cross-validation resulted in a sensitivity of 0.6701, a specificity of 0.6815, and an MCC of 0.3515. For iSNO-AAPair [130], 2300 nitrosylated sites and 2300 non-nitrosylated sites from dbSNO (only from human and mouse data) were selected as part of the training set, and 81 nitrosylated and 100 non-nitrosylated were selected as part of the testing set (none of which were in the training set). Ten-fold cross-validation over 50 iterations revealed an average of 0.852 for sensitivity, 0.790 for specificity and 0.64 for MCC. For PSNO [134], the training set was constructed from dbSNO, identical to the subset derived in iSNO-PseAAC. PSNO was then compared to GPS-SNO, iSNO-PseAAC and iSNO-AAPair using the independent dataset that benchmarked iSNO-AAPair’s performance. PSNO had the highest MCC of 0.72 against an MCC of 0.28 for GPS-SNO, an MCC of iSNO-PseAAC of 0.30, and an iSNO-AAPair MCC of 0.63.

The computational analysis of the surrounding environment that would dictate the electronic resonance structure and thus dictate SNO stability has been suggested by Talipov and Timerghazin [136]. They propose that there are three resonance structures in SNO formation. The first structure termed “D structure” with a short double bond between the sulfur and nitrogen atoms, positively charged S and negatively charged O. In this structure, the S–N bond is stronger, than its resonant opposite, “I structure”. In this structure, there exists an ion pair between the S and N atoms, creating a longer and weaker S–N bond, with a positive charge on triple-bonded N–O and a negative charge on the S atom. The “S structure” is an intermediate of the two forms. Analysis by a variety of computational methods of the protein environment, nitrosating agent and steric permissibility will dictate which of these structures is favored and this will account for the feasibility and stability of the formed nitrosothiol bond. Further development and expansion of this method in tangent with verified motifs may prove to be promising in the prediction of SNO formation.

## 6. Mechanisms of SNO Formation

Several general chemical mechanisms by which an SNO can be formed have been proposed [4,5]. In order to appreciate the contribution of a chemical mechanism to selectivity, the potential reactions of NO and thiol must be considered. As stable SNO formation requires a formal single electron oxidation, formation reactions can be broadly subdivided based on when in the process of reaction between the NO donor and the thiol the oxidation occurs: (1) pre-oxidation: either NO or SH undergoes one electron abstraction prior to reaction; (2) concurrent: change in the metal ion oxidation state during reaction of thiol and NO; and (3) post-oxidation: involves electron abstraction following the direct reaction of NO and SH moieties.

### 6.1. Pre-Oxidation

Several mechanisms by which the NO moiety may undergo one electron oxidation to form the nitrosonium cation, NO^+^, are possible. It should be noted that NO^+^ readily reacts with water to form NO_2_^−^ in aqueous environments and thus would not be capable of nitrosation in itself. For effective nitrosation of cysteines, a stable NO^+^ donor must exist. The ability of high NO concentrations, particularly in hydrophobic environments, to generate the nitrosating agent N_2_O_3_ through autooxidation may be a mechanism for RSNO formation, with the NO_2_^−^ leaving group effectively removing the additional electron [137]. Perhaps the most well-established mechanism for SNO formation is transnitrosation from one SNO to another cysteine thiol, with dinitrosyl iron complexes participating in this mechanism where an NO^+^ is transferred to an acceptor thiol [103]. *S*-nitrosoglutathione (GSNO) often participates as the NO^+^ donor due to its high intracellular abundance [138]. In this mechanism, NO^+^ transfer is governed by the chemical equilibrium between GSNO and the protein thiols available for transnitrosation. NO^+^ will preferentially be transferred to and retained on thiols where SNO formation is thermodynamically favored. In addition to the differential affinity of the thiol, the ability of SNO–thiol pairs to physically interact is essential for these reactions [139]. The small size of glutathione may explain, at least in part, its ability to participate in many transnitrosation reactions. The presence of glutathione binding sites has been shown to be important in facilitating transnitrosation. For larger proteins to be effective NO^+^ donors, close interaction between recipient and donor thiols is required, and this proximity may be enhanced by specific tertiary structural features. Such reactions have been observed for certain donor proteins such as thioredoxin [140], ceruloplasmin [141] and GAPDH [142].

One electron oxidation of thiolate anion will produce a thiyl radical capable of direct reaction with NO via doublet–doublet interaction. Such oxidation may occur as a result of oxidative/nitrosative stress [112,143,144], metal/metalloprotein–thiol interactions [145,146,147], enzymatic catalysis or impaired antioxidant response. Hydrogen peroxide has been shown to oxidize thiolate anion, while peroxynitrite preferentially oxidizes protonated thiols at physiologic pH [148], catalyzed via the presence of CO_2_ [149]. NO production may indirectly aid in producing thiyl radical, as NO_2_ is capable of one electron oxidation of thiol, generating NO_2_^−^. The activity of endogenous antioxidant systems, such as the glutathione/glutathione reductase and thioredoxin/thioredoxin reductase, modifies thiol redox state, with variable activity in certain cell types or physiologic contexts influencing the likelihood and targeting of SNO formation.

### 6.2. Concurrent Oxidation

Some redox active metals have been shown to facilitate SNO formation in solution, but their incorporation into proteins with specific enzymatic activities is required for specific targeted thiol nitrosylation. SNO formation will depend on the element in question, metal coordination within the protein structure, metal oxidation state, and a geometry sufficient to accommodate both the NO donor and the protein thiol. Metals coordinated within these enzyme structures can bind with NO radicals, stabilize an intermediate form, often through a change in metal oxidation state and facilitate electron abstraction by coordination of potential acceptors. Though enzyme structure will encourage or restrict the proximity of metal-bound NO to specific thiols, the coordinated metal may also influence SNO reaction rate. Notably, the extraordinary affinity of copper for thiols [150] allows for the efficient transfer of NO^+^ from metal to S^−^ [151]. These interactions form the basis for a major potential mechanism of SNO synthesis, which is via cytochrome c [152]. In known cases of metal-catalyzed SNO formation, the metalloenzyme typically abstracts the radical electron during transfer to the thiol, effectively performing a transnitrosation. Experimental evidence also supports a role of chelatable redox-active copper [153] and iron [103] in the formation of transnitrosating species. 

Two examples of SNO forming enzymes demonstrating the diversity of metal catalysis in SNO formation are ceruloplasmin and hemoglobin. It has been demonstrated that the cysteine thiol in glutathione is nitrosylatable via the action of ceruloplasmin. In the proposed mechanism of catalysis by ceruloplasmin one of six Cu^2+^ molecules binds NO, changing its oxidation state to Cu^+^ [93]. The step at which glutathione enters the catalytic site is unclear and may be before following or consequent to NO–Cu binding. It is also possible that the reacting glutathione is initially bound to a proximate copper rather than the NO-bound metal. In either case, effective nitrosation occurs as the ceruloplasmin retains the electron following the disassociation of GSNO. It has been proposed that the coordination of several proximate copper molecules allows for the sharing of the electron prior to donation to a suitable physiologic acceptor. Hemoglobin provides a microcosm of NO biology, as it is capable of nitrosylation, nitrosation, nitration and nitroxylation reactions [154]. For SNO formation, when NO is bound to the T state heme, iron encounters molecular oxygen and forms a peroxynitrite-like intermediate. Molecular rearrangement upon conformational change from T to R state results in the iron bonding oxygen, and the interaction between the nitrogen and the βCys93, which occurs simultaneously with the proceeding step [97]. Cleavage of the O–N bond occurs with O_2_ acting as the electron acceptor, leaving as superoxide and restoring the iron to the 2^+^ oxidation state. A stable SNO is formed at the βCys93, which can be donated to other proteins or glutathione via transnitrosation [155]. 

### 6.3. Post-Oxidation 

Direct reaction between NO radical and protein thiol groups has been suggested as a plausible means for the production of SNO [156]. Such reaction between free cysteine amino acid thiols and NO radical have been shown to occur in experimental conditions [157]. It has been proposed that attack on the S–H bond produces an RSNOH radical intermediate via a rapidly reversible reaction [158]. When an electron acceptor is present to remove the additional electron, deprotonation gives the stable SNO. If the electron is not abstracted, the reverse reaction occurs and the SNO is not formed. In an aerobic environment, molecular oxygen can act as the electron acceptor, resulting in the formation of a stable SNO and superoxide. Under anaerobic conditions, reduced dinucleotides, such as NAD^+^, can act as the acceptor [157]. Though only demonstrated in vitro for free cysteine molecules, it is conceivable that within the context of proteins, such reactions can occur. Specific tertiary structures of proteins can potentially increase the rate of oxidation and the specificity of the intermediate by increasing the proximity of the reactive thiol to electron acceptors. 

It has been proposed that certain circumstances may exist whereby the formation of the RSNOH radical, though inherently unstable, is sufficiently active to produce a physiologic effect prior to electron extraction and that electron removal is involved in the rapid termination of the signal [159]. Work by Kolesnik et al. further validates the possibility of a direct reaction between radical NO and a thiol using GSH through the elimination of the formation of peroxynitrite and the use of low concentrations of NO, eliminating autooxidation in experimental conditions. Though unable to demonstrate NAD^+^ as an electron acceptor, they found that such a reaction is first order, linearly dependent only on NO, yielding a 100% efficiency of nitrosation when NO concentrations are low and GSH concentration is high. These studies point to a direct reaction of NO with thiol that is dependent upon the chemistry of the thiol involved.

## 7. Mechanisms of SNO Degradation

Just as the production of SNO proteins is a highly regulated process, the removal of NO moieties from proteins is physiologically regulated. Though principally regarded as a mechanism by which SNO signaling is terminated, several proteins—notably the caspases 3 [29] and 9 [23] and matrix metalloprotinase-9 [160]—have been demonstrated to be constitutively inactivated by *S*-nitrosylation and are only activated by the removal of NO. In aortic endothelium, self-nitrosylation of eNOS has been demonstrated to tonically inhibit its own activity, with physiologic denitrosylation subsequent to VEGF or insulin signaling, transiently permitting enzymatic function [161]. It has been speculated that, in addition to terminating physiologic signals, the enzymatic denitrosylation of proteins modified during conditions of nitrosative stress plays a key role in the termination of pathological adaptations and the restoration of the physiologic state.

Either heterolytic or homolytic S–N bond cleavage can occur, releasing either NO^+^, NO^.^ or NO^−^ [162,163]. In this context, either NO^+^ or NO radicals are able to react with other nearby thiols (with NO^+^ being the predominantly produced species), while NO^−^ has been shown to accelerate disulfide bond formation [162,164] or generate peroxynitrite. In cell-free systems, the rate and decay products of low molecular weight and protein SNOs has been found to be pH dependent, with homolytic cleavage occurring within physiologic pH ranges, while heterolytic cleavage was detected at pH higher than 9.0 [165,166,167]. These data also suggest that the pKa of the thiol may also have significant impact on the rates of S–N bond cleavage. Homolytic S–N bond cleavage has been demonstrated in studies of bacteria exposed to low molecular weight SNOs [168], as well as in mammalian cell culture systems, where pancreatic cells stimulated with acetylcholine produce NO in a calcium-dependent manner [169]. Broadly, the mechanisms by which SNO proteins undergo denitrosylation can be classified as either non-enzymatic or enzymatic in nature.

### 7.1. Non-Enzymatic Decomposition

The simplest form of SNO decomposition is homolytic cleavage induced by UV light, which has been proposed as the mechanism of NO release in the skin [170]. The photolytic hemolysis of the S–N bond results in the formation of free NO and thiyl radical [171]. Often, the thiyl group can undergo radical recombination resulting in disulfide bond formation. Photolytic decomposition may be involved in the pathogenesis of melanoma development [172,173]. As patients with psoriasis have increased levels of NO within psoriatic lesions owing to increased expression of iNOS [174], it is conceivable that a portion of patient responsiveness to phototherapy involves the decomposition of pathologically nitrosylated proteins. 

Metal catalyzed decomposition is possibly the most common form of SNO decomposition observed in biology. The presence of trace transition metals in solution has been observed to dramatically increase the rate of SNO decomposition. Notably, the presence of free copper [149,175], mercury [176] and selenium [177] in solution are likely to contribute to the acceleration of SNO decomposition. Reduced metals, particularly Cu(I), are potent catalysts for SNO decay [151]. Ascorbate and glutathione may enhance SNO decomposition by maintaining metals in their reduced form [178]. In general, the free concentrations of redox metals are low as they are sequestered by proteins, and therefore, these reactions are more relevant to metalloproteins. 

SNOs can be degraded by oxidative decomposition in vivo, although the relevancy of these reactions may be limited. At physiologic concentrations of oxygen, SNO is unlikely to undergo autooxidation; however, N_2_O_3_ has been proposed as a potential decomposition accelerant [179]. Nitrosonium ion has been proposed to initiate cyclic decomposition of SNOs through development of RSNOSR^+^ intermediates that react with other SNOs to release 2NO, forming an unstable RSNOSR^+^ [163]. These intermediates are capable of reaction with other SNOs, forming stable disulfide pairs and regenerating the reactive RSNONO+ for chain propagation. Chain termination may occur by NO transfer from RSNOSR^+^ to non-thiol nucleophiles. Under conditions of oxidative stress, SNO decomposition via reaction with superoxide causes the release of peroxynitrite and disulfide formation [180]. 

### 7.2. Enzymatic Decomposition 

As mentioned, metals fixed within proteins have the potential to participate in SNO chemistry. Fixing metals within a particular protein structure both imparts substrate specificity and influences the kinetics of SNO decomposition. In the case of copper, over 95% of the mammalian physiologic stores are bound to ceruloplasmin, a known SNO synthase [93]. Copper is also found in association with serum albumin and metallothioniens; currently, however, the role of these copper proteins in SNO biology is unknown. A number of other metal containing enzymes, such as SOD1 [181], the bacterial and yeast flavohemoglobins [182,183], and the selenoproteins glutathione peroxidase and thioredoxin reductase [184,185,186,187,188] have been shown to have the capacity to act as denitrosylases.

The modification of the size and distribution of the intracellular SNO pool is an essential function of GSNO reductase enzyme (GSNOR), a class III alcohol dehydrogenase enzyme whose SNO metabolizing functions are highly conserved [189]. The in vitro characterization of its activity revealed the selective decomposition of GSNO to a glutathione N-hydroxysulphenamide product without the direct catalytic effect on protein SNOs. This reaction consumes NADH and has been demonstrated to be irreversible [190], with the GSNHOH product likely restored to the free glutathione pool via glutathione reductase, producing ammonia or hydroxylamine in the process. GSNOR-knockout mice have increased levels of both low-molecular weight and protein SNOs, including some SNO proteins not demonstrated to be nitrosylated under physiologic conditions [191]. These GSNOR knock out mice have been demonstrated to have a propensity for the development of hepatic tumors, while human hepatocellular carcinoma patients have diminished GSNOR activity and expanded SNO protein pools [192]. In GSNOR-deficient mice, ineffective denitrosylation was shown to result in increased proteasomal degradation of the DNA repair enzyme O6-alklguanine DNA alkyltransferase, providing a potential explanation for both spontaneous and toxin-induced carcinoma development [192].

Enzymes that alter the cellular redox environment by the generation of reactive oxygen species can accelerate the decomposition of SNOs. SNO decomposition via Xanthine Oxidase involves superoxide-mediated reduction of R-SNO to yield NO radical [193]. This process can be inhibited by Cu/Zn SOD, suggesting that a hydroxyl radical is considerably less effective in facilitating denitrosylation. In anerobic conditions, the reaction of xanthine and Cys-SNO in the presence of xanthine oxidase resulted in uric acid production and NO radical liberation. Pathologic denitrosylation as a result of xanthine oxidase activity has been demonstrated to contribute to arythmogenesis and organ failure in the rat heart.

The regulation of thiol redox status is also associated with the dentirosylation processes. The thioredoxin/thioredoxin reductase system (Trx/TrxR) plays an essential role in disulfide reduction and protection from oxidative stress. Trx contains a highly conserved Cys–Gly–Pro–Cys motif [194], placing vicinal thiols within the catalytic site and facilitating the decomposition of RSNO to RSH, releasing HNO. Several molecular mechanisms of SNO decomposition to HNO have been proposed. The consensus mechanism suggests that the nucleophilic vicinal thiol pair protonates both the NO moiety and thiol through an undetermined reaction intermediate that stabilizes HNO as a leaving group and leaves Trx with an intramolecular disulfide bond [25,195]. Alternate mechanisms have been proposed whereby transient covalent protein–Trx intermediates are formed [195] or where the transnitrosation of Trx occurs and is released by autocatalysis [196]. In either mechanism, the net result is the restoration of HNO, denitrosylated protein, and oxidized Trx with intramolecular disulfide. This disulfide is reduced by TrxR using NADPH as an electron donor. It has also been suggested that Trx is capable of decomposing GSNO, with a resultant shift in SNO equilibrium similar to that seen in the case of GSNOR [197]. As three distinct molecular mechanisms with different structural intermediates have been proposed, it is conceivable that Trx may facilitate denitrosylation through different mechanisms dependent on structural aspects of the SNO.

## 8. Conclusions

The addition or removal of NO from protein sulfhydryl groups is an essential mechanism for redox-sensitive cellular signaling, conserved from ancient microbiota throughout modern eukaryotic life. *S*-nitrosylation is as fundamental a form of posttranslational modification as phosphorylation. The likelihood of any cysteine thiol being modified is determined by the chemical reactivity of NO, the local biological milieu and the redox state of the thiol pool. Ultimately this means that there are individual pools of SNOs that are generated by different mechanisms and are subjected to degradation by a variety of means. Thus, one can say that not all SNOs are created equal, and it becomes critical to determine the factors that control which thiols are modified. Broadly speaking, there are three main factors that determine which thiols are susceptible to modification; (1) the flux rate of NO production; (2) the local concentration of reactants (such as metals and oxidants); and (3) the microenvironment of the thiol moiety itself. The local flux rate of NO is controlled by its rate of production, the diffusion rate and the rate of reaction with potential targets. The local concentration of reactants is largely determined by the intracellular redox status, which is a major contributing factor in both the availability of redox active metals as well as the formation of oxidants. The proclivity of residues to undergo stable modification is a function of the thiol microenvironment, including coordinated metal proximity, glutathione docking and the stability of thiolate anion. As the pool of SNOs is dynamic, the reversal of nitrosylation is equally important in determining the nature of that pool as the formation reactions. The reversal of nitrosylation may be achieved through enzymatic and non-enzymatic mechanisms, although there is much work still to be done to fully understand these processes. A greater knowledge of how SNO equilibria are regulated will be critical in understanding the physiologic implications of these modifications.

## Figures and Tables

**Figure 1 antioxidants-10-01111-f001:**
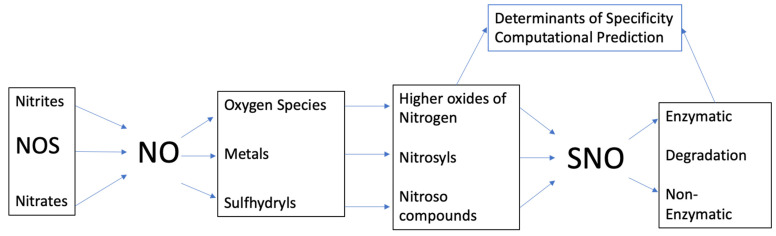
Scheme of areas to be covered in this review. This review will cover the literature of how SNOs are formed with consideration of the sources of NO and the reactants available within the biological milieu and the mechanisms of degradation. There is a focus on how these factors lead to the specificity of SNO formation and how predictive modeling may allow for the determination of sites of SNO formation.

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
