# Peer review of "Biological Mechanisms of S-Nitrosothiol Formation and Degradation: How Is Specificity of S-Nitrosylation Achieved?"

_antioxidants, 2021, doi:10.3390/antiox10071111_

Round 1

Reviewer 1 Report

Post-translational modification of cysteine residues by NO, creating S-Nitrosothiols, is considered an important signaling mechanism in the cell. This review is timely, necessary, and comprehensive, seeking to elaborate on the formation and degradation of S-Nitrosothiols. With a few minor adjustments, the manuscript could be improved:

  1. Among the sources of NO, are there data to indicate which sources contribute most to the S-Nitrosylation, especially compared to others? For instance, are the listed dietary sources a physiologically relevant contributor to the process?
  2. If it were feasible, a graphic outlining the major themes (e.g., sources, formation and degradation) would be very helpful.
  3. Please elaborate on the redox potential of NO and how it contributes to electron transfers in the cell.

Author Response

We thank the reviewer for their comments and have altered the manuscript in line with their suggestions.

1)  We have added to the sources of NO section a statement about the work of the Winyard group which recently demonstrated that ingestion of beets can increase vascular SNOs

2) We have added a graphic to summarize the areas covered by the review and how they connect.

3) We have added references to the Bartberger and Fukuto papers that discuss the redox active potential of NO

Reviewer 2 Report

In the paper "Biological Mechanisms of S-Nitrosothiol Formation and Degra-2 dation: How is Specificity of S-Nitrosylation Achieved?" the biological chemistry of NO and nitrosothiol formation is reviwed

The paper is  well written and organized

Minor point

  1. Please add a paragraph on  phisological effetcs of NO
  2. Please add a paragraph on role of NO in cardiac disease

Author Response

We thank the reviewer for their comments.  The paper is focused on the cysteine modifications mediated by NO (partly as this a submission to a special issue on redox regulation of cysteines).  Therefore we didnt think it was reasonable to add whole paragraphs on the physiological functions of NO in general adn the cardiovascular system in particular.  However, the reviewer's comments and are well made in that positioning the paper within the wider context may be helpful.  Therefore, we have edited the opening paragraph to incorporate statements about NO function and to give references to papers that explore these topics more fully.